# Metatranscriptomic Analysis of Oropharyngeal Samples Reveals Common Respiratory Viruses and a Potential Interspecies Transmitted Picobirnavirus in the Wayuu Population, La Guajira, Colombia

**DOI:** 10.3390/v17101397

**Published:** 2025-10-21

**Authors:** Beatriz Elena De arco-Rodríguez, Jhindy Tatiana Pérez-Lozada, Katherine Laiton-Donato, Dioselina Peláez-Carvajal, Gloria Mercedes Puerto-Castro, Diego Alejandro Álvarez-Díaz

**Affiliations:** 1Grupo de Genómica de Microorganismos Emergentes, Dirección de Investigación en Salud Pública, Instituto Nacional de Salud, Bogotá 111321, Colombia; bedea@unal.edu.co (B.E.D.a.-R.); jhperezl@unal.edu.co (J.T.P.-L.); klaiton@ins.gov.co (K.L.-D.); dpelaez@ins.gov.co (D.P.-C.); 2Grupo de Micobacterias, Dirección de Investigación en Salud Pública, Instituto Nacional de Salud, Bogotá 111321, Colombia; gpuerto@ins.gov.co; 3Grupo de Investigación y Desarrollo de Vacunas y Biológicos Estratégicos en Salud Pública, Dirección de Producción, Instituto Nacional de Salud, Bogotá 111321, Colombia

**Keywords:** viruses, metatranscriptomic, acute respiratory infection, Wayuu indigenous, Colombia

## Abstract

Acute respiratory infections and other infectious diseases causing acute febrile syndrome are major public health concerns in Colombia, particularly among vulnerable populations such as the Wayuu Indigenous community in Manaure, La Guajira. To investigate their viral etiology, 55 nasopharyngeal swabs and 58 serum samples were collected from febrile Wayuu individuals in Manaure. RT-qPCR screening identified Coronavirus, Enteroviruses, Adenovirus, and Influenza A/B in respiratory samples, while no arboviruses were detected in serum. Sixteen representative samples underwent metatranscriptomic next-generation sequencing (mtNGS) using the Chan-Zuckerberg ID (CZ-ID) platform. This analysis confirmed RT-qPCR findings and additionally revealed six viral contigs related to *Orthopicobirnavirus hominis*. Sequencing coverage enabled the reconstruction of a consensus RdRp segment, which was phylogenetically compared with sequences from diverse hosts. The virus clustered within genogroup 1, alongside Colombian isolates linked to severe acute respiratory infection. The absence of strict host-specific clustering suggests possible interspecies transmission. These findings underscore the complementary roles of targeted and unbiased approaches: RT-qPCR detected common respiratory viruses, whereas mtNGS uncovered a virus previously unreported in this community. Overall, mtNGS emerges as a powerful tool to support viral surveillance and provide baseline evidence in indigenous populations, emphasizing the need to decentralize advanced molecular diagnostics and strengthen public health capacity in Colombia.

## 1. Introduction

In Colombia, acute respiratory infections (ARI) and unknown-etiology diseases are significant causes of acute febrile syndrome, which encompasses a wide spectrum of microbial agents and represents one of the leading reasons for seeking medical attention [1,2]. This burden is particularly relevant in tropical and rural regions, where zoonotic and emerging viral diseases are more prevalent [3,4]. Despite advances in health surveillance, underreporting of infectious diseases persists, especially in vulnerable communities such as indigenous ethnic groups, delaying accurate diagnosis and timely clinical management; for this report, we will focus on the Wayúu people of the Manaure savannah in La Guajira, Colombia.

The Wayuu ethnic group faces common disparities in low-income communities that result in adverse consequences on health, mainly related to substandard housing, including a lack of safe drinking water, improper waste disposal, food insecurity, vector-borne diseases, and living in hard-to-reach areas that compromise healthcare access [5]. All of this increases the vulnerability of the Wayuu people in the Manaure savannah, where the number of ARI and febrile syndrome notifications in emergency rooms and outpatient care services increases on an annual basis, creating a significant public health challenge [6,7].

To address these challenges, total RNA-based metatranscriptomic analysis was performed to enable the unbiased identification of pathogenic or potentially pathogenic microorganisms that remain undetected by conventional diagnostic methods, as well as the detection of new and emerging viruses. This approach could be useful for supporting epidemiological surveillance, outbreak control, and management strategies for infectious diseases in Colombia.

## 2. Materials and Methods

### 2.1. Participants, Samples and Study Area

In November 2023, sampling was conducted under the program “Formulation of a Strategy for the Prevention, Control, and Management of Communicable Diseases with a One Health Approach in Manaure, La Guajira”. From a Wayuu population belonging to the indigenous communities of Ichien and Polousira (Figure 1), and displaying clinical signs of acute respiratory infection and fever, with or without an apparent infectious focus, and after obtaining informed consent and authorization from the indigenous authority, we collected 58 serum samples and 55 nasopharyngeal swabs. The samples were transported under a cold chain to the Laboratorio de Genómica de Microorganismos Emergentes at the Instituto Nacional de Salud in Bogotá, Colombia.

### 2.2. RNA Purification and RT-qPCR for Virus Detection

Total RNA was purified from the nasopharyngeal swab and serum samples using the ExiPrep™ Viral DNA/RNA kit on the ExiPrep™48 Dx system (Bioneer Corporation, Daejeon, Republic of Korea), following the manufacturer’s instructions. The RNA was quantified by fluorometry using the Qubit™ RNA Broad Range (BR) Assay Kit (Life Technologies, Carlsbad, CA, USA) on a Qubit 4.0™ instrument (Life Technologies, Carlsbad, USA).

RT-qPCR on RNA extracts to verify the Ct value for *Betacoronavirus pandemicum* (formerly severe acute respiratory syndrome coronavirus 2-SARS-CoV-2), *Alphainfluenzavirus influenzae* and *Betainfluenzavirus influenzae* (formerly Influenza A and B), *Alphacoronavirus amsterdamense* (formerly Human coronavirus NL63-HCoV-NL63), *Betacoronavirus hongkongense* (formerly Human coronavirus-HKU1 HCoV-HKU1), *Alphacoronavirus chicagoense* (formerly Human coronavirus 229E-HCoV-229e), Metapneumovirus, *Orthopneumovirus hominis* (formerly Respiratory Syncytial Virus), Adenovirus, *Enterovirus betacoxsackie* (formerly Enterovirus B), and *Enterovirus alpharhino* (formerly Rhinovirus A) [8,9,10,11,12,13,14,15,16]. Following metatranscriptomic detection, an additional RT-qPCR assay targeting the capsid gene of Orthopicobirnavirus was conducted for molecular confirmation, using the primer set described by Berg et al. [17]. The viral taxa screened or detected in this study, along with their corresponding families and genome types, are summarized in Table 1.

For RNA extracts from serum samples, RT-qPCR assays were conducted for arboviruses: Dengue virus (DENV), Zika virus (ZIKV), and Chikungunya virus (CHIKV) [18,19].

### 2.3. Metatranscriptomic Sequencing

Metatranscriptomic sequencing was performed to capture the actively transcribed fraction of viral diversity, with particular focus on RNA viruses. Libraries were prepared using the automated MGISP-100 system (MGI Tech, Shenzhen, China). To enrich non-host transcripts, mammalian ribosomal RNA (rRNA) was removed with the MGIEasy rRNA Depletion Kit. The depletion process involved hybridization of rRNA to specific capture probes, followed by RNase H-mediated digestion. Residual DNA was subsequently removed using DNase 1 treatment, and the remaining target RNA was purified with magnetic beads. The depleted and purified RNA was fragmented and converted to complementary DNA (cDNA) via reverse transcription with random primers, followed by second-strand synthesis. The resulting double-stranded cDNA underwent end-repair, A-tailing, adapter ligation and PCR enrichment according to the manufacturer’s RNA-seq Library Prep Kit Protocol (MGI Tech, Shenzhen, China). PCR products were subsequently purified using magnetic beads. Library concentration was quantified using the Qubit™ dsDNA High Sensitivity (HS) Assay Kit (Life Technologies, Carlsbad, CA, USA), and quality was assessed by evaluating the expected fragment size with D1000 DNA ScreenTape assays on a TapeStation 4150 system (Agilent Technologies, Santa Clara, CA, USA).

All individual libraries were pooled in equimolar amounts and subsequently circularized for DNA nanoball (DNB) synthesis. DNBs were generated from the single-stranded DNA circles obtained during library preparation and used as templates for sequencing. Library concentration was maintained at ≥2 fmol/μL, with each DNB reaction requiring 40 fmol of library in a 100 μL reaction volume. A minimum DNB concentration of 12 ng/μL was required for sequencing. Finally, sequencing was performed on a DNBSEQ-G50 platform (MGI Tech, Shenzhen, China) using a single-end 100 bp (SE100) FCL flow cell with 120 sequencing cycles.

### 2.4. Bioinformatics Analysis

Raw sequencing reads were quality-filtered and processed using the CZ-ID pipeline v8.3 [20]. CZ-ID is an open-source, cloud platform used to identify microbial agents in NGS datasets. This pipeline integrates multiple bioinformatic tools into an automated workflow that enables researchers to analyze sequencing data without requiring local computational infrastructure. Specifically, in this study the pipeline performed the following steps: Fastp for removal of short and low-quality reads, CZID-dedup to collapse PCR/optical duplicates without affecting transcript diversity, Bowtie for the initial subtraction of host reads, and Hisat2 for the removal of remaining host reads. The remaining reads were aligned against the NCBI nucleotide (NT) and non-redundant protein (NR) databases (index date: 6 February 2024) using Minimap2 and DIAMOND, respectively, and preliminary taxonomic assignments were generated. Reads passing filters were assembled into contigs with SPAdes, followed by refinement through BLAST-based (v2.16.0) reassignments against candidate NT and NR references, improving taxonomic accuracy. The final outputs included annotated contigs, refined taxon counts, coverage statistics, and non-host FASTQ files for downstream analyses.

To determine the presence of viral taxa in each sample, five filters were applied to generate the heatmap to compare the relative presence of viral taxa across samples: 1. Category: non-phage viruses; 2. NT rPM ≥ 20 (number of reads aligning to a taxon in the NCBI NT database, per million reads sequenced); 3. NT alignment length ≥ 50 base pairs; 4. NR rPM ≥ 1 (number of reads aligning to a taxon in the NCBI NR protein database); and 5. Z-score filter > 1, based on a standard background model, to retain only taxa more abundant in samples than in negative controls. NT rPM was used to estimate virus abundance in each sample.

### 2.5. Phylogenetic Tree of RdRp from Orthopicobirnavirus Hominis

A consensus sequence of segment 2 (encoding the RNA-dependent RNA polymerase, RdRp) was generated by mapping reads and contigs to a reference RdRp sequence using BWA-MEM 0.7.17-r1188, with SAMtools 1.13 for processing and masking of low-coverage regions with Ns. To evaluate the reliability and completeness of the reconstructed consensus, a coverage plot was done using the ggplot2 package in R 4.5.1.

A nucleotide dataset comprising complete and partial sequences of segment 2 (RdRp) of Picobirnaviruses (PBV) was assembled, including the Colombian sequence from this study, previously reported sequences from the same country, and additional representatives from GenBank (accessed 10 July 2025). Sequences were translated to aminoacid using the correct open reading frame, and multiple sequence alignments were performed at the protein level to account for codon redundancy and improve homology assessment. The dataset covered diverse host species and geographic origins and included representatives from the three major RdRp phylogeny-based genogroups: PBV1, PBV2, and PBV3 (Appendix A).

Multiple sequence alignment (MSA) was performed with MAFFT v7 using the E-INS-i algorithm. All ambiguously aligned regions were removed using trimAl v1.4, as implemented in Knox et al. [21], due to the high sequence divergence of PBV that generates extensive gaps in the alignment. The trimAl software itself was originally described by Capella-Gutiérrez et al. [22]. The best-fit protein substitution model (LG + I + G4) was selected using ModelFinder in IQ-TREE v2.2.0.3 based on the Bayesian Information Criterion (BIC). Phylogenetic reconstruction was performed with IQ-TREE employing the selected model and 10,000 ultrafast bootstrap replicates. The resulting tree was midpoint-rooted and visualized with iTOL v7.4.

### 2.6. Ethical Considerations

The study was conducted following the Declaration of Helsinki. All subjects enrolled in this study voluntarily signed and informed consent form previously approved by Comité de Ética y Metodologías de Investigación (CEMIN 18-2023), and prior authorization was obtained from the indigenous leader.

## 3. Results

A total of 69 participants were included in the study, 39.1% (*n* = 27) male and 60.9% (*n* = 42) female. The mean age was 12.9 years, ranging from 0.8 to 62 years. Most of the individuals were between the ages of 5 and 14 years. A total of 52.2% (*n* = 36) reported fever, and 43.5% (*n* = 30) presented with cold/flu-like symptoms (Table 2).

From a total of nasopharyngeal swabs (*n* = 55), 9.1% were positive by RT-qPCR for *Alphacoronavirus amsterdamense*, 10.9% for *Enterovirus alpharhino*, 5.45% for Influenza A/B, 5.45% for *B. hongkongense*, 1.82% for *Betacoronavirus pandemicum*, 1.82% for Adenovirus and 1.82% for *Enterovirus betacoxsackie*. Regarding the serum samples (*n* = 58), no arboviruses were detected by qPCR. A subset of 13 swab (COL_INS_GME_01 to COL_INS_GME_13) and 3 serum (COL_INS_GME_14 to COL_INS_GME_16) samples met the required concentration (>10 ng/μL) for metatranscriptomic sequencing. Results are shown in Table 3.

The viral taxon classification was based on CZ-ID pathogen lists, tagging them as “virus non-phage.” Thresholds for classifying taxa based on the number of reads are shown in Figure 2, which illustrates the adjusted parameters used for more accurate classification.

On average, 22,002,253 sequence reads were generated per sample, of which 3,320,099 passed filtering. Viral sequences were detected in five oropharyngeal swabs by mtNGS analysis, including *Alphacoronavirus amsterdamense*, *Enterovirus alpharino* and *Enterovirus betacoxsackie* (Figure 2). Overall, the findings were consistent with PCR results, except for sample 05. No viral reads were detected in serum samples.

Notably, two distinct viral taxa were identified in the oropharyngeal swab sample COL_INS_GME_06: *Betainfluenzavirus influenzae* and *Orthopicobirnavirus hominis* (Figure 1). Six contigs related to *Orthopicobirnavirus hominis* recovered from CZ-ID were compared against both databases of NCBI, BLASTn and BLASTx. In both searches, three contigs aligned with 94% and 92% identity to capsid sequences (e.g., OL875325.1 by BLASTn and ULB12850.1 by BLASTx, respectively), while the remaining three aligned with RNA-dependent RNA polymerase (RdRp) sequences (e.g., OL875336.1 by BLASTn and UYF11801.1 by BLASTx) of *Orthopicobirnavirus hominis* sequences in Genbank, supporting their viral origin and taxonomic classification as a Human PBV.

The mtNGS allowed the recovery of near-complete coverage with 93.6% for segment 1 (capsid) and partial coverage with 91.2% for segment 2 (RdRp) of *Orthopicobirnavirus hominis.* Although the overall sequencing depth was limited, the contigs generated were sufficient to support downstream phylogenetic analyses (Appendix A).

To further confirm this, an RT-qPCR targeting the capsid segment of *Orthopicobirnavirus hominis* was performed using primers and a FAM-labeled probe described by Berg et al. [17]. No amplification was observed, likely due to the assay’s specificity for a particular strain. An RNAse P RT-PCR served as an internal control and confirmed RNA integrity and assay performance (Appendix A).

Following the genomic identification of *Orthopicobirnavirus hominis*, a phylogenetic analysis was conducted using a 1421 bp consensus sequence of the RdRp segment. The sequence was compared with sequences of the three genogroups (PBV1, PBV2, and PBV3) proposed for the PBV genus, based on the genetic variability of segment 2, as outlined by Perez et al. [23]. The resulting tree showed a clear separation into genogroups, with PBV1 (blue) and PBV3 (green) clustered more closely together, while genogroup 2 (orange) formed a distinct and more divergent branch (Figure 3A). Sequences from genogroup 1 were the most abundant and dispersed across multiple subclades. No consistent clustering by host species or geographic origin was observed. The RdRp sequence from the Colombian sample COL_INS_GME_006 clustered within the genogroup 1 and formed a strongly supported clade with previously reported Colombian sequences hosted by *Homo sapiens* (e.g., OL875336.1) (Figure 3B). This clade also included additional Colombian sequences of PBV derived from respiratory samples [23,24].

## 4. Discussion

Although nucleic acid detection by qPCR is the gold standard for viral diagnosis, it is limited to preidentified viral genomic sequences [25]. By contrast, the mtNGS approach generates millions of unbiased reads, enabling virus detection without prior sequence knowledge. In this study, nearly all viruses identified by RT-qPCR were also detected by sequencing, underscoring the robustness of mtNGS as a high-throughput approach for simultaneously assessing viral diversity across multiple samples and identifying pathogens at the species level in Wayuu individuals presenting ARI [2,26]. Beyond serving as an exploratory tool, mtNGS provides an opportunity to uncover previously unrecognized viruses circulating in Colombia, strengthening viral surveillance and public health efforts against respiratory diseases [17,27].

In particular, the identification of *Alphacoronavirus amsterdamense* in COL_INS_GME_03, *Enterovirus alpharhino* in COL_INS_GME_04 and 07, and *Enterovirus betacoxsackie* in COL_INS_GME_08 was consistent between RT-qPCR and mtNGS techniques. Moreover, the CZ-ID tool revealed high genomic coverage from mtNGS data, with 79% for *Enterovirus alpharhino* and 99.7% for *Enterovirus betacoxsackie* B4, demonstrating the reliability of this approach for detecting specific viruses. This concordance aligns with previous studies where CZ-ID has proven effective for pathogen identification and genome recovery directly from clinical and environmental samples [28,29,30]. Conversely, the relatively low coverage (4.4) observed for *Alphacoronavirus amsterdamense* likely reflects low viral load, as suggested by its Ct value. These findings underscore the ability of mtNGS to detect viral genomes, including those present at relatively low abundance, as supported by the concordance observed with qPCR results [31].

Viruses identified here have been reported in respiratory disease outbreaks, particularly among children, the elderly, and immunocompromised patients. Consistent with global epidemiological trends, the majority of study participants were Wayuu children aged 5–14 years, who are especially vulnerable to respiratory viral infections due to biological susceptibility, limited access to healthcare, environmental exposures, and communal living conditions [32,33]. *Alphacoronavirus amsterdamense* is associated with upper and lower respiratory tract infections and is recognized as an important cause of laryngotracheobronchitis (croup) [34,35,36]. Although its surveillance in Colombia remains limited, it is globally distributed and occasionally linked to severe cases in vulnerable populations [34]. Similarly, enteroviruses, associated with the common cold, are highly prevalent in tropical regions such as Colombia, where they may progress to severe disease requiring intensive care and acute respiratory distress syndrome. National surveillance data from late 2023 reported rhinoviruses and enteroviruses as contributors to 15.7% and 11.5% of severe respiratory cases, respectively [37].

A key finding of this study was the detection of *Orthopicobirnavirus hominis*. While its capsid gene was not amplified by qPCR, mtNGS successfully recovered both capsid and RdRp segments. The discrepancy is likely attributable to the primer set employed, which was originally designed to detect a specific viral strain [17]. The absence of capsid amplification reflects primer-template mismatch introduced by sequence divergence at binding sites, rather than by a true lack of this genomic segment. Consistently, Berg et al. [17], reported respiratory samples that were capsid-negative but RdRp positive, suggesting circulation of capsid-divergent PBV strains. Notably, our RdRp sequence clustered with one of these capsid-negative samples (OL875336) [17]. These findings underscore the limitations of strain-specific assays when applied to genetically diverse viral groups such as PBV. Although targeted qPCR validation of the RdRp would be an ideal next step, this was not performed in the present study. The assembly of contigs covering both the capsid and RdRp segments, together with their high-identity alignment to reference sequences in public databases, provides strong molecular evidence for the presence of the virus. In addition, our consensus RdRp sequence was aligned against previously published degenerate primers designed to amplify all known PBV [17], showing correct matches to these primers. In that context, the recent application of metatranscriptomic shotgun sequencing has enabled the detection of highly divergent or previously uncharacterized RNA viruses [17,38].

Despite the relatively low sequencing depth for *Orthopicobirnavirus hominis*, the coverage achieved across both genomic segments provides strong evidence of viral presence in the analyzed sample (Appendix A). This outcome is common in metatranscriptomic studies, where viral reads typically constitute only a minor fraction of the total dataset, with host and bacterial transcripts predominating [39,40]. Similar findings in other studies emphasize the utility of mtNGS to identify unexpected viral agents even under low read abundance [29,41].

The co-detection of *Orthopicobirnavirus hominis* and *Betainfluenzavirus influenzae* suggests a potential involvement of both viruses in respiratory illness. Phylogenetically, our *Orthopicobirnavirus hominis* sequence branched with a subset of viral sequences from sputum samples of Colombian patients hospitalized with ARI, underscoring its potential association with severe respiratory disease [17]. This represents the second report of the circulation of this virus in Colombia, although from a different region than the one originally described. Moreover, our phylogenetic analysis of the RdRp gene supported previous relationships between some Colombian sequences and respiratory-derived sequences from China and Cambodia, suggesting persistent phylogenetic links among lineages associated with respiratory samples across geographically distant regions [17]. Moreover, previous studies have also identified PBV in hospitalized patients with severe acute respiratory infection of probable zoonotic origin and in patients with respiratory illness of unknown etiology [42,43]. Despite its frequent detection in stool samples, similar clustering patterns involving respiratory-associated sequences were also observed by Pérez et al. [23].

Phylogenetic analysis of the RdRp also supported established divergence among PBV genogroups (PBV 1-PBV 3), with PBV 2 remaining the most distant lineage, as previously reported by Perez et al. [23]. PBV is one of the many novel RNA viruses recently discovered in several animal hosts through mtNGS. Furthermore, PBV sequences from different animal species were observed to be distributed throughout the phylogenetic tree. Sadiq et al. [38] reported a lack of host-specific clustering, highlighting the virus’s capacity to cross species boundaries [38]. The absence of a consistent host-associated phylogenetic pattern has challenged the classification of viral species according to their host [19]. Future research could focus on examining the genetic diversity of this virus as it remains unclear whether it is an animal or prokaryotic virus, and its association with respiratory symptoms to shed light on the pathogenicity, transmission, and potential public health impact of this virus in Colombia [23,38].

Finally, no viruses were detected in serum samples from Wayuu febrile participants, which may indicate that the underlying etiology was not viral, potentially involving bacterial, parasitic, or even non-infectious causes. To advance pathogen surveillance in remote regions, future studies should include a larger number of samples and implement nucleic acid stabilization systems during field collection and transport, thereby improving the detection and characterization of infectious agents. Furthermore, by focusing solely on viruses, the analysis overlooks other pathogenic microbes; a broader, microbiome-based approach is therefore needed to fully understand the causes of febrile illnesses in these populations.

## 5. Conclusions

This study shows that mtNGS is a valuable tool for detecting emerging and previously uncharacterized pathogens in northern Colombia. In the Wayuu population, we identified respiratory viruses and reported the country’s second case of *Orthopicobirnavirus hominis*, a virus that may contribute to severe respiratory disease. Expanding this approach with representative sampling could strengthen surveillance and guide public health policies for vulnerable communities. Importantly, these efforts also help build local diagnostic capacity and support the decentralization of health services, demonstrating how the INS can collaborate with regional health systems to extend access to advanced technologies for pathogen surveillance [38,44].

## Figures and Tables

**Figure 1 viruses-17-01397-f001:**
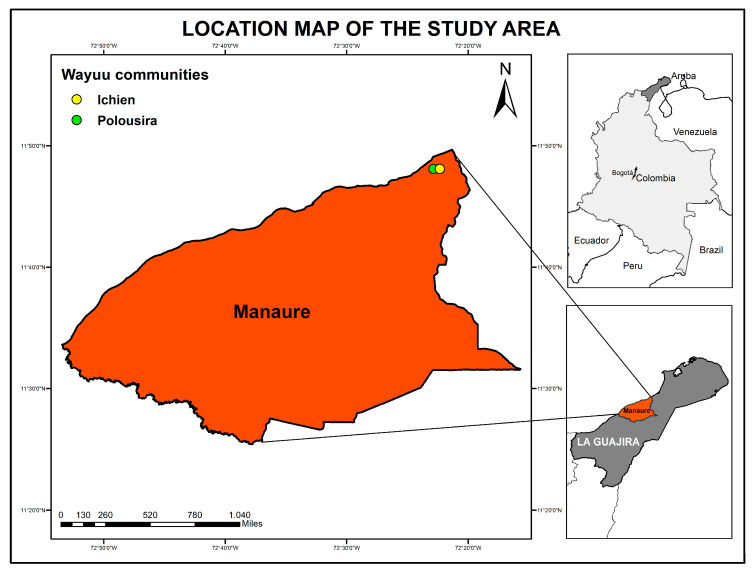
Location map of the study area. The map shows Colombia with the department of La Guajira shaded in dark gray. A zoomed view of the municipality of Manaure, shown in terracotta, highlights the Indigenous communities of Ichien (yellow) and Polousira (green). For national context, the location of Bogotá is also indicated. The map was created using ArcGIS v10.6.

**Figure 2 viruses-17-01397-f002:**
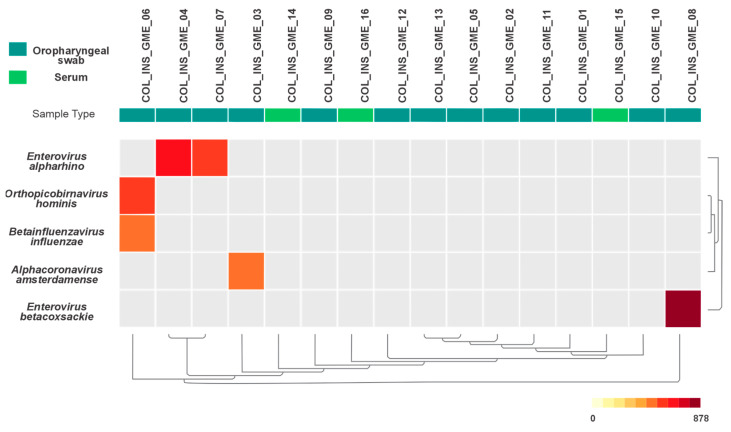
Viral profile overview. Heatmap displaying the relative abundance of viral taxa detected by CZ-ID mNGS pipeline. Gray cells indicate taxa not found in the sample or filtered out during analysis.

**Figure 3 viruses-17-01397-f003:**
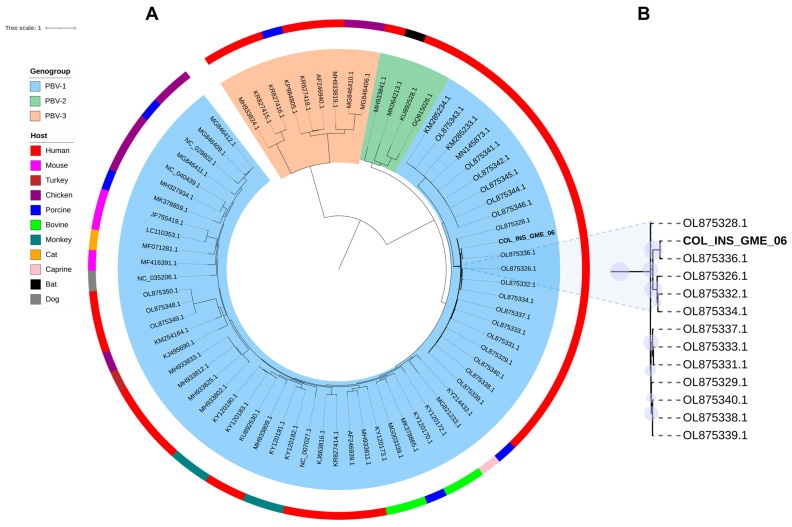
Maximum-likelihood phylogeny of PBV based on RdRp protein sequences. (**A**) Tree including representative sequences from genogroups 1 in block shaded (blue), 2 (orange), and 3 (green), with host information indicated. (**B**) Enlarged view of the clade containing the Colombian index case (COL_INS-GME-06) highlighted in black. Bootstrap values >70% are shown as light blue circles at key nodes. Alignment was generated with MAFFT (E-INS-i), and the best-fit model (LG + I + G4) was selected in IQ-TREE. Branch lengths represent substitutions per site.

**Table 1 viruses-17-01397-t001:** Respiratory viruses screened in this study, showing their corresponding taxonomic classification and genomic characteristics.

Virus Name	Common/Formerly Name	Family/Genus	Genome Type
*Betacoronavirus pandemicum*	SARS-CoV-2	Coronaviridae/*Betacoronavirus*	ssRNA (+)
*Alphainfluenzavirus influenzae*	Influenza A	Orthomyxoviridae/*Alphainfluenzavirus*	ssRNA (−), segmented
*Betainfluenzavirus influenzae*	Influenza B	Orthomyxoviridae/*Betainfluenzavirus*	ssRNA (−), segmented
*Alphacoronavirus amsterdamense*	HCoV-NL63	Coronaviridae/*Alphacoronavirus*	ssRNA (+)
*Betacoronavirus hongkongense*	HCoV-HKU1	Coronaviridae/*Betacoronavirus*	ssRNA (+)
*Alphacoronavirus chicagoense*	HCoV-229E	Coronaviridae/*Alphacoronavirus*	ssRNA (+)
*Orthopneumovirus hominis*	Respiratory Syncytial Virus (RSV)	Pneumoviridae/*Orthopneumovirus*	ssRNA (−)
*Metapneumovirus*	Human Metapneumovirus	Pneumoviridae/*Metapneumovirus*	ssRNA (−)
Adenovirus	Human adenovirus	Adenoviridae/*Mastadenovirus*	dsDNA
*Enterovirus betacoxsackie*	Coxsackievirus (Enterovirus B)	Picornaviridae/*Enterovirus*	ssRNA (+)
*Enterovirus alpharhino*	Rhinovirus A	Picornaviridae/*Enterovirus*	ssRNA (+)
*Orthopicobirnavirus hominis*	Human Picobirnavirus	Picobirnaviridae/*Orthopicobirnavirus*	dsRNA, segmented

**Note:** The abbreviations ss and ds refer to single- and double-stranded genomes, whereas the symbols (+) and (−) indicate the positive- and negative-sense RNA or DNA genomes, respectively.

**Table 2 viruses-17-01397-t002:** Sociodemographic and clinical characteristics of the participants.

Category	Description	Number	Percentage
Gender	Male	27	39.1
	Female	42	60.9
	Total	69	100
Age	<5 years	23	33.3
	5–14 years	27	39.1
	15–29 years	10	14.5
	30–59 years	8	11.6
	≥60 years	1	1.5
	Total	69	100
Symptoms *	Fever	36	52.2
	Cold/Flu-like symptoms	30	43.5
	Diarrhea	22	31.9
	Cough	13	18.8
	Stomach pain	4	5.8
	Headache	3	4.3
	Sore throat	3	4.3
	Vomiting	2	2.9
	Nausea	1	1.4
	Ear pain	1	1.4
	Skin rash/spots	1	1.4
	Kidney pain	1	1.4
	Loss of appetite	1	1.4
	None/No symptoms	8	11.6

* Multiple symptoms were reported by some participants.

**Table 3 viruses-17-01397-t003:** Results of virus detection by RT-qPCR and RNA sequencing.

Sample Name	Type of Sample	RT-qPCR		Metatranscriptomic	
		Ct	Virus	Reads Per Million	Virus
COL_INS_GME_01	nasopharyngeal swab	-	Negative	-	Negative
COL_INS_GME_02	nasopharyngeal swab	-	Negative	-	Negative
COL_INS_GME_03	nasopharyngeal swab	34.14	*Alphacoronavirus* *amsterdamense*	31.3	*Alphacoronavirus amsterdamense*
COL_INS_GME_04	nasopharyngeal swab	30.53	*Enterovirus alpharino*	129.4	*Enterovirus alpharino*
COL_INS_GME_05	nasopharyngeal swab	29.64	*Enterovirus alpharino*	-	Negative
COL_INS_GME_06	nasopharyngeal swab	29.74	Influenza A/B	21.91	*Betainfluenzavirus influenzae*
				70.48	*Orthopicobirnavirus hominis*
COL_INS_GME_07	nasopharyngeal swab	28.85	*Enterovirus* *alpharino*	75.24	*Enterovirus alpharino*
COL_INS_GME_08	nasopharyngeal swab	25.64	*Enterovirus* sp.	878	*Enterovirus* *betacoxsackie*
COL_INS_GME_09	nasopharyngeal swab	-	Negative	-	Negative
COL_INS_GME_10	nasopharyngeal swab	-	Negative	-	Negative
COL_INS_GME_11	nasopharyngeal swab	-	Negative	-	Negative
COL_INS_GME_12	nasopharyngeal swab	-	Negative	-	Negative
COL_INS_GME_13	nasopharyngeal swab	-	Negative	-	Negative
COL_INS_GME_14	Serum	-	Negative	-	Negative
COL_INS_GME_15	Serum	-	Negative	-	Negative
COL_INS_GME_16	Serum	-	Negative	-	Negative

**Note:** The symbol “-” indicates no detection: no Ct value for RT-and no reads for metatranscriptomics.

## Data Availability

Sequencing data are openly available in NCBI under BioProject PRJNA1336963, BioSample SAMN52096369, and SRA accession SRR35697695 (accessed on 1 October 2025).

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
