# Peer review of "Metatranscriptomic Analysis of Oropharyngeal Samples Reveals Common Respiratory Viruses and a Potential Interspecies Transmitted Picobirnavirus in the Wayuu Population, La Guajira, Colombia"

_viruses, 2025, doi:10.3390/v17101397_

Round 1
Reviewer 1 Report
Comments and Suggestions for Authors
Summary: Metatranscriptomic Analysis of Oropharyngeal Samples Reveals Common Respiratory Viruses and a Potential Interspecies Transmitted Picobirnavirus in the Wayuu Population, La Guajira, Colombia by De arco-Rodríguez takes an RT-qPCR and agnostic metatranscriptomic sequencing dual approach to investigate the etiology of respiratory disease among the Wayuu people of Colombia. The authors examined 55 nasopharyngeal swabs and 58 serum samples collected from febrile individuals and identified several common respiratory viruses including coronaviruses, enteroviruses, adenoviruses and influenza viruses. The authors also detected the presence of human Picornobirnavirus (Orthopicobirnavirus hominis), a little studied virus found in multiple mammalian hosts and potentially linked to respiratory and/or enteric infectious disease in humans.
Strengths of the Article: The study is important from the public health perspective by focusing research resources on the largest indigenous community in Colombia, who have limited access to healthcare, disease surveillance and diagnostic capacity. The application of both RT-qPCR and metatranscriptomic sequencing to the study is a powerful use of these technologies for studying disease etiology. The detection of Orthopicobirnavirus hominis adds to the very limited literature on the subject of global human Picobirnavirus distribution and disease etiology.
Limitations:
- Relatively small sample size for metatranscriptomic analysis (16 total samples)
- Only one sample yielded Orthipicobirnavirus hominis. This sample had low sequencing depth coverage and was not detected by RT-qPCR due to primer failure.
Revisions:
- The authors should design new primers based on the metatranscriptomic sequence data to confirm the detection of Orthopicobirnavirus hominis in sample COL_INS_GME_06 to confirm that the lack of detection was due to primer failure and not nucleic acid integrity due to storage and transport from a remote location.
- Minor:
- Line 41: “emcompases a wide sprectrum of microbial agent” → “encompasses a wide spectrum of microbial agents”.
- Line 66: “fever or fever without an apparent infectious focus” → “fever with or without an apparent infectious focus”. Also, not clear that focus is the correct word choice in this sentence.
- Line 205: “Maximun-likelihood phylogeny” → “Maximum-likelihood phylogeny”.
- Line 243:” Consistently, previous studies have also identified Picobirnavirus in hospitalized patients with severe acute respiratory infection of probable zoonotic origin and in patients with respiratory illness of unknown etiology”. The word consistently does not fit here. Use a phrase such as “Consistent with the hypothesis that..” or “Moreover…
- References: Ensure consistent formatting (uniform DOI citation style, remove redundant “[Internet]” notations).
Author Response
Thank you for taking the time to review our manuscript. We sincerely appreciate the reviewers’ constructive feedback. Detailed responses to all comments are provided below, and the suggested changes have been incorporated into the revised version of the manuscript, where they are underlined in the re-submitted file.
Limitations:
Relatively small sample size for metatranscriptomic analysis (16 total samples). Only one sample yielded Orthipicobirnavirus hominis. This sample had low sequencing depth coverage and was not detected by RT-qPCR due to primer failure.
Comment: We acknowledge the limitation of having a relatively small sample size and that only one sample yielded the mentioned virus with low sequencing depth and not detected by RT-qPCR. Nonetheless, this finding highlights the sensitivity of the metatranscriptomic approach in detecting viral RNA even when present at low abundance, and underscores the challenges of primer-based diagnostics, which may miss divergent or low-titer viruses. The identification of a diverse set of viral taxa, despite the sample size, provides valuable baseline data on the respiratory virome of this vulnerable indigenous population and supports the utility of mNGS for pathogen detection in underrepresented regions. In the revised version, we have explicitly mentioned this limitation and indicated that future studies including a larger number of samples will be conducted to validate and expand upon these findings. See page 11, line 354-357.
Revisions:
- The authors should design new primers based on the metatranscriptomic sequence data to confirm the detection of Orthopicobirnavirus hominis in sample COL_INS_GME_06 to confirm that the lack of detection was due to primer failure and not nucleic acid integrity due to storage and transport from a remote location.
Comment: We appreciate the observation regarding the need to validate the Orthohopicobirnavirus hominis finding via qPCR. The detection of virus in sample COL_INS_GME_06 was unexpected and highlights the value of unbiased metatranscriptomic sequencing. While qPCR validation was not been performed in the present study, the finding is supported by strong molecular evidence: the assembly of significant contigs and their consistent alignment with reference sequences in public databases. To further assess the robustness of this result, we aligned our consensus RdRp sequence against previously published degenerate primers designed to amplify all known Picobirnaviruses (Berg et al. 2021), and the alignment indicated correct matching to these primers. We included a statement in the Discussion noting that on page 10, line 309-315, and is highlighted in yellow in the manuscript. The alignment file of the consensus RdRp sequence with the published degenerate primers will be included as supplementary material for the editor’s review. Additionally, we have made the assembled consensus sequence and raw reads publicly available (BioProject PRJNA1336963; BioSample SAMN52096369; SRA SRR35697695), enabling independent assessment of the data. Indeed, the automated analysis provided by NCBI (including taxonomic profiling visualized through a Krona plot) also identified Orthopicobirnavirus hominis together with Betainfluenzavirus influenzae in sample COL_INS_GME_06, independently confirming our findings.
- Minor:
- Line 41: “emcompases a wide sprectrum of microbial agent” → “encompasses a wide spectrum of microbial agents”.
Comment 1: Thank you for pointing this out. The typographical error has been corrected; the test now reads [which encompasses a wide spectrum of microbial agents and represents one of the leading reasons for seeking medical attention]. See Page 2, line 42-43.
- Line 66: “fever or fever without an apparent infectious focus” → “fever with or without an apparent infectious focus”. Also, not clear that focus is the correct word choice in this sentence.
Comment 2: Thank you for pointing this out. The typographical error has been corrected; the test now reads [acute respiratory infection and fever, with or without an apparent infectious focus]. See Page 2, line 70.
- Line 205: “Maximun-likelihood phylogeny” → “Maximum-likelihood phylogeny”.
Comment 3: Thank You for pointing this out. The typographical error has been corrected; the test now reads [Maximum-likelihood phylogeny of Picobirnavirus]. See Page 9, line 255.
- Line 243:” Consistently, previous studies have also identified Picobirnavirus in hospitalized patients with severe acute respiratory infection of probable zoonotic origin and in patients with respiratory illness of unknown etiology”. The word consistently does not fit here. Use a phrase such as “Consistent with the hypothesis that..” or “Moreover…
Comment 4: Thank you for pointing this out. The typographical error has been corrected; the test now reads [Moreover, previous studies have also identified Picobirnavirus in hospitalized patients]. See Page 10, line 335.
- References: Ensure consistent formatting (uniform DOI citation style, remove redundant “[Internet]” notations).
Comment 5: Thank you for pointing this out. The typographical error has been corrected; the test now reads without “[Internet]” notations.
Reviewer 2 Report
Comments and Suggestions for Authors
In this article De arco-Rodrìguez and colleagues provide updated information on the research progress regarding the use of mtNGS technology for detecting emerging and previously uncharacterized pathogens, including those with potential for interspecies transmission. The main objective is to provide tools that strengthen public health in Columbia, but it can also be taken as an example in other areas of the world.
The article is well written; bibliography is updated, however, there are some points that need to be reviewed. I have several detailed suggestions:
Query 1. Some families of viruses are mentioned that are probably not known to everyone. It would be advisable to add a figure or table showing the main structural characteristics of these viruses.
Query 2. Paragraph 2.1 Participants and samples: A map of Columbia showing the location of the region studied would be helpful for those unfamiliar with the area (you can make a figure showing the map of Columbia with the study area (sampling area- La Guajira- and the city of Bogotà).
Query 3. Paragraph 2.2 RNA purification and RT-qPCR for virus detection: To compare the performances of RT-qPCR and mtNGS, it would be appropriate to indicate the Limit of Detection (LOD) and the cut-off of the RT-qPCR method for the different targets analyzed. This is partly because the authors state (line 232): “These findings highlight the capacity of mtNGS to recover viral genomes even at low abundance”.
Query 4. Paragraph 2.3 Metatranscriptomic sequencing: Since mtNGS constitutes the central element of the study, it is advisable to provide a more comprehensive account of the experimental workflow used.
Query 5. Paragraph 2.4 Bioinformatics analysis: It would be advisable to expand and clarify the description of the CZID platform, as it may not be well known, especially to those unfamiliar with the work.
Query 6. Line 134: “picobirnaviruses” has already been appointed in line 124; it can be replaced with the abbreviation.
Query 7. In accordance with the comment on line 234, it would be appropriate to include a statement regarding the age of the patients in this study, noting that most of them are between 5 and 14 years old.
Query 8. Table 1: Check the sum: 33,3 + 39,1 + 14,5 + 11,6 + 1,4 does not equal 100. Accordingly, the rounding of the figures should be reviewed.
Query 9. Table 1: It is necessary to review the sum in the symptoms section. Since the sum of the percentages does not add up to 100, the 100% total leads to misunderstanding. In my personal opinion, I would remove the total from this part of the table.
Query 10. Line 254: Check the spelling of the word Picobirnaviruses. However, it can be replaced with the abbreviation.
Query 11. Line 274: Picobirnaviruses can be replaced with the abbreviation.
Query 12. Line 281: The parentheses can be removed, and Picobirnaviruses can be replaced with the abbreviation.
Author Response
Thank you for taking the time to review our manuscript. We sincerely appreciate the reviewers’ constructive feedback. Detailed responses to all comments are provided below, and the suggested changes have been incorporated into the revised version of the manuscript, where they are underlined in the re-submitted file. In addition, adjustments were made to the Introduction to improve clarity and strengthen the contextual background. These modifications are highlighted in yellow in the re-submitted manuscript.
Query 1. Some families of viruses are mentioned that are probably not known to everyone. It would be advisable to add a figure or table showing the main structural characteristics of these viruses. Adjustments were made to the Introduction section to improve clarity and contextual relevance. These modifications are highlighted in yellow in the re-submitted manuscript.
Comment 1: Thank you for this valuable suggestion. We agree that the taxonomic names used in the manuscript may not be familiar to all readers. To address this, we have added a new table (Table 1) after the paragraph describing the RT-qPCR screening. This table summarizes the viruses detected in this study, including their current ICTV name, common/previous name, family, genome type, which we believe will facilitate reader understanding. See page 3, lines 100-103.
Query 2. Paragraph 2.1 Participants and samples: A map of Columbia showing the location of the region studied would be helpful for those unfamiliar with the area (you can make a figure showing the map of Columbia with the study area (sampling area- La Guajira- and the city of Bogotà).
Comment 2: Thank you for this useful suggestion. Following the recommendation, we have modified the title of Section 2.1 to “Participants, samples, and study region”. In this section, we now include a figure (figure 1) showing a map of Colombia, highlighting the department of La Guajira, the municipality of Manaure, and the two indigenous communities that participated in this study, as well as the capital city of Bogotá for reference. See page 2, lines 65-79.
Query 3. Paragraph 2.2 RNA purification and RT-qPCR for virus detection: To compare the performances of RT-qPCR and mtNGS, it would be appropriate to indicate the Limit of Detection (LOD) and the cut-off of the RT-qPCR method for the different targets analyzed. This is partly because the authors state (line 232): “These findings highlight the capacity of mtNGS to recover viral genomes even at low abundance”.
Comment 3: We appreciate the reviewer’s insightful comment regarding the importance of indicating the Limit of Detection (LOD) and the cut-off values for the RT-qPCR assays. We agree that such parameters are essential when conducting comparative analytical evaluations. However, our study did not aim to perform a quantitative comparison between RT-qPCR and mtNGS, but rather to characterize the respiratory virome within the study population. RT-qPCR was applied to verify the presence of common respiratory viruses, supporting the overall concordance of detection with mtNGS data. The sentence has been revised to better reflect this purpose on page 9, lines 281-283.
Query 4. Paragraph 2.3 Metatranscriptomic sequencing: Since mtNGS constitutes the central element of the study, it is advisable to provide a more comprehensive account of the experimental workflow used.
Comment 4: Thank you for your observation. Section 23 has been expanded in order to provide a more detailed description of the metatranscriptomic workflow.These modifications have been highlighted in yellow in the manuscript. See Paragraph 2.3 on page 4.
Query 5. Paragraph 2.4 Bioinformatics analysis: It would be advisable to expand and clarify the description of the CZID platform, as it may not be well known, especially to those unfamiliar with the work.
Comment 5: Thank you for your valuable suggestion. The description of the CZ-ID pipeline has been expanded in the Bioinformatics analysis section to provide greater clarity for readers unfamiliar with the platform. In the revised text, we now detail the main steps of the pipeline (quality filtering, host subtraction, duplicate removal, alignment to NT/NR, contig assembly, and refinement), specify the NCBI index date used (2024-02-06), and clarify that the final filtered results were visualized in a heatmap to compare the relative presence of viral taxa across samples. It should be noted that CZ-ID will transition to the University of California, San Francisco’s Institute for Global Health Sciences (IGHS) toward the end of 2025. These modifications have been highlighted in yellow in the manuscript. See Paragraph 2.4.
Query 6. Line 134: “picobirnaviruses” has already been appointed in line 124; it can be replaced with the abbreviation.
Comment 6: Thank you for pointing this out. The abbreviation has been added; the test now reads PBV. See Page 5, line 173.
Query 7. In accordance with the comment on line 234, it would be appropriate to include a statement regarding the age of the patients in this study, noting that most of them are between 5 and 14 years old.
Comment 7: We appreciate your review. A statement has been added in the Results section (Page 5, line 188), immediately before the demographic table, indicating that most patients in this study were between 5 and 14 years old. Additionally, we incorporated this age-related observation into the Discussion section, where it complements the epidemiological context of respiratory viral susceptibility in children, as suggested. It is highlighted in yellow in the manuscript. see Page 9, lines 286-290.
Query 8. Table 1: Check the sum: 33,3 + 39,1 + 14,5 + 11,6 + 1,4 does not equal 100. Accordingly, the rounding of the figures should be reviewed.
Comment 8: Thank you for noticing this. The error stemmed from an improper rounding of the number 1.49 to 1.5. The table has now been corrected. See Page 6.
Query 9. Table 1: It is necessary to review the sum in the symptoms section. Since the sum of the percentages does not add up to 100, the 100% total leads to misunderstanding. In my personal opinion, I would remove the total from this part of the table.
Comment 9: We appreciate your review. The table has now been corrected. See Page 6.
Query 10. Line 254: Check the spelling of the word Picobirnaviruses. However, it can be replaced with the abbreviation.
Comment 10: Thank you for noticing this. The abbreviation has been added; the test now reads PBV.
Query 11. Line 274: Picobirnaviruses can be replaced with the abbreviation.
Comment 11: Thank you for noticing this. The abbreviation has been added; the test now reads PBV. This correction has been applied consistently throughout the manuscript.
Query 12. Line 281: The parentheses can be removed, and Picobirnaviruses can be replaced with the abbreviation.
Comment 12: Thank you for noticing. This change have been incorporated on line 335.